# Iron Dysregulation and Frailty Syndrome

**DOI:** 10.3390/jcm10235596

**Published:** 2021-11-28

**Authors:** Bartosz Zawadzki, Grzegorz Mazur, Aleksandra Butrym

**Affiliations:** 1Department and Clinic of Internal and Occupational Diseases, Hypertension and Clinical Oncology, Wrocław Medical University, 50-367 Wroclaw, Poland; grzegorz.mazur@umw.edu.pl; 2Department of Cancer Prevention and Therapy, Wrocław Medical University, 50-367 Wroclaw, Poland

**Keywords:** iron regulation, nutrition, hospitalized patients

## Abstract

Patients with diagnosed frailty syndrome (FS) represent a special group of patients with chronic disease. In the classic definition, frailty syndrome includes such parameters as reduced muscle strength, subjective feeling of fatigue, unintentional weight loss, slow gait, and low physical activity. Frailty syndrome leads to an increased incidence of adverse events, such as falls, hospitalizations, and the need to place patients in care and health institutions associated with the loss of independence; frailty syndrome is also associated with an increased incidence of death. In European countries, the frequency of frailty syndrome in the geriatric population is estimated to be 17% with a range from 5.8% to 27%, and its incidence increases with age. A much higher percentage of frailty syndrome patients is also observed among hospitalized patients. The incidence of frailty syndrome is influenced by many socio-economic factors, but also medical factors. Materials and Methods: A total of 120 patients, >65 years of age, participated in the study. During the study, anthropometric measurements, surveys, laboratory determinations of basic biochemical parameters, and iron status were investigated; 5 mL of peripheral blood in EDTA was also collected for further laboratory tests of hepcidin and soluable transferrin receptor (sTfR) using ELISA. Then, the statistical analysis was performed based on survey and clinical data. Results: Among the patients >65 years of age, the incidence of frailty syndrome was 27.5%. It was found that its occurrence was associated with socio-economic factors, malnutrition, multiple morbidities, reduced muscle strength and gait speed, and polypharmacotherapy. The relationship between reduced iron concentration and the occurrence of frailty syndrome was confirmed. Conclusions: According to the analysis, it was found that a decrease in iron concentration was associated with frailty syndrome.

## 1. Introduction

People in Poland aged >65 years old can expect to live less than half their lives in health (men 7.5 years, i.e., 47% and women 8.1 years, i.e., 40%) [1]. Polish patients were treated most often for cardiovascular diseases (15% of hospitalized); injuries and poisonings (9.2% of hospitalized); cancers (9.2% of hospitalized); as well as urogenital, digestive, and respiratory diseases (7.8%, 7.6%, 6.8% of hospitalized, respectively). Cardiovascular diseases were responsible for 45.1% of total deaths and malignant neoplasms were the second most common cause of death in Poland (25.4% of all deaths in 2014) [1]. Patients with diagnosed frailty syndrome (FS) are a special group of hospitalized patients with chronic disease. It is extremely difficult to clearly define this complex geriatric problem due to its interdisciplinary nature. According to the definition proposed by L. P. Fried [2], it is a physiological syndrome characterized by a decrease in reserves and resistance to stressors. In the classic definition, frailty syndrome includes such parameters as reduced muscle strength (hand grip below 20% for the norm established for sex and body mass index (BMI)), subjective feeling of fatigue, unintentional weight loss (at least 4.5 kg per year), slowed gait (below 20% of the norm established for sex, measured as the time to cover a distance of 4.572 m), and low physical activity. Three of the abovementioned criteria are sufficient for diagnosis. Frailty syndrome is usually preceded by prefrailty syndrome, which includes occurrence of one to two of the symptoms and identifies a group of elderly at a significantly increased risk of developing frailty syndrome. Frailty results from the accumulation of reduced efficiency of various physiological systems over the course of life, which leads to susceptibility to adverse effects [3]. Frailty syndrome leads to an increased incidence of adverse events such as falls, hospitalizations, and institutionalizations, which are associated with loss of independence and affect increased mortality [4]. FS is an independent predictor of mortality, even after considering co-existing chronic diseases and other factors [5]. In particular, in the older adults >65 years old, iron deficiency anemia, which is one of the most frequent health problems in older adults, is most closely associated with an increase in IL-6 [6] and low-grade active inflammation [7]. Anemia is characterized by some combination of iron restriction, insufficient bone marrow response, and decreased red blood cell survival [8]. There is increasing evidence that the aging process affects iron metabolism [9]. Identifying iron deficiency has become an increasing problem due to age-related changes in hemoglobin, the effects of prescribed medications on age-related disorders and diseases, and increased levels of inflammation-related ferritin and hepcidin [10,11]. According to the World Health Organization (WHO), approximately 25% of the world’s population suffer from iron deficiency anemia [12,13]. Anemia of chronic diseases, which includes the presence of low-intensity chronic inflammation, for example, in rheumatic, cancer and infectious diseases [14] might be associated with frailty syndrome as well. A comprehensive assessment of iron metabolism is possible thanks to the study of the transferrin-receptor system for transferrin and determination of the concentration of the protein that binds iron in cells, i.e., ferritin [15]. Recently, much attention has also been paid to the protein regulating iron metabolism, i.e., hepcidin, since it has been recognized as a potential indirect indicator of iron stores in the body [16]. Iron dysregulation seems to be an important link in the complex problem of developing frailty syndrome. 

## 2. Materials and Methods

This study involved 120 patients hospitalized in the Department of Internal, Occupational Diseases, Hypertension, and Clinical Oncology of the University Clinical Hospital in Wroclaw, Poland. Each patient gave written consent to participate in the study and was informed about the purpose of the study, its course, and the possibility of withdrawing at each stage of the study. The criteria for inclusion in the study were: age >65 years, health status enabling informed consent to participate in the study, and written consent to participate in the study. The first stage of the study included conducting surveys, which included: the Edmonton Frailty Scale (EFS) [17], the Mini Nutritional Assessment (MNA) [18], the Nutritional Risk Score (NRS) [19], and clinical surveys covering basic sociodemographic data (date of birth, place of residence, education, and social conditions) and medical (medications taken, chronic diseases, hospitalizations, dietary supplements taken, blood transfusions, and used medical supplies). In case of doubt regarding a patient’s cognitive abilities, the MMSE (Mini-Mental State Examination) test was performed. A score of >21 points qualified the patient for further participation in the study. In addition, anthropometric measurements, such as height, weight, BMI, waist circumference, hip circumference, arterial pressure, heart rate, and muscle strength, as well as laboratory tests were performed. The Bioethics Committee at the Medical University in Wroclaw gave consent to carry out the research included in the project. The procedures complied with the ethical standards included in the 2013 Helsinki Declaration. The analysis of qualitative variables (i.e., not expressed in numbers) was performed by calculating the number and percentage of occurrences of each value. The comparison of quantitative variable values in two groups was performed using the Mann–Whitney test. Comparison in three or more groups was made using the Kruskal–Wallis test and when statistically significant differences were detected, a post hoc analysis was performed using the Dunn test to identify statistically significantly different groups. Correlations between quantitative variables were analyzed using the Spearman correlation coefficient. The level of significance in the analysis was 0.05; therefore, all *p*-values below 0.05 were interpreted as indicating significant relationships. The analysis was performed in the R program, version 3.6.1. [20].

## 3. Results

### 3.1. Group Characteristics

The study group included 49 men and 71 women. The average age of the patients was 74.6 years (men 74.1 years and women 74.9 years) (Figure 1). For the majority, i.e., 55 patients, it was their first hospitalization in a year, for 35 patients it was their second hospitalization in a year, for 11 patients it was their third hospitalization in a year, and 19 patients had been hospitalized more than 3 times in the previous year. The most common causes of hospitalization included: hypertension, anemia, and exacerbation of chronic heart failure (Table 1). The sociodemographic data is presented in table (Table 2).

### 3.2. Edmonton Frailty Scale Results

The incidence of frailty syndrome in patients was set at 27.5%. Sixty-three patients (52.5%) were non-frail, 24 (20%) patients were in the risk group, and 33 (27.5%) patients had the frailty syndrome. The level of frailty was light in 16 patients, medium in 11 patients, and severe in 6 patients (Table 3). Age is one of the stronger risk factors for frailty syndrome. It has been shown that the older the patient population, the more frequent the frailty syndrome. In the study group, age correlated significantly (*p* ˂ 0.05) and positively (r ˃ 0) with the total score on the Edmonton Scale and with such subscales as: cognition, functional independence, continence and functional performance. However, no significant relationship with the gender of the respondents was found in the study population. Another socio-economic domain that was studied was level of education. The total score on the Edmonton Scale was significantly higher among people with primary and secondary education than among people with higher education. The influence of place of residence on the correlation with weakness was also examined as an indirect indicator of the socio-economic situation. Higher points on the frailty scale were found among inhabitants of large cities than among inhabitants of villages and small towns.

### 3.3. Clinical Markers of Frailty

It was found that malnutrition described on the MNA and the NRS scale was significantly associated with frailty syndrome. Higher waist-hip ratio (WHR) and higher BMI, higher muscle strength correlated negatively with the Edmonton Scale. Cigarette smoking (daily use) and alcohol consumption (>40 g/week) were not significantly associated with frailty. Significantly higher total scores on the Edmonton Scale were obtained in groups of patients hospitalized for exacerbation of chronic heart failure and anemia. Lower concentrations of albumin and total protein, as well as lower levels of low-density lipoprotein (LDL), total cholesterol, high-density lipoprotein (HDL), and magnesium were observed in frail patients. Glucose levels were significantly lower in patients without frailty. It was found that patients with reduced serum iron concentration had a significantly higher risk of frailty syndrome. Other basic laboratory parameters were also tested in patients with frailty syndrome. In this way, characteristic changes and deviations in biochemical parameters and blood counts were identified. Peripheral blood counts showed significantly lower levels of hemoglobin, hematocrit, and lymphocytes, and higher levels of neutrophils as compared with other elderly patients. Another important element was the description and occurrence of inflammation, which, currently, has been postulated as the basic pathomechanism of frailty syndrome development. In the study group, C-reactive protein (CRP) was used, as well as, indirectly, the concentration of ferritin. Inflammation parameters (CRPs) significantly and positively correlated with the total Edmonton Score, especially in the areas of cognitive ability, functional independence, and functional performance.

### 3.4. Iron Status and Frailty Syndrome

The key element of this study was the assessment and analysis of the dependence of iron parameters. Laboratory parameters such as iron concentration, total iron binding capacity (TIBC), unsaturated iron binding capacity (UIBC), ferritin, soluble transferrin receptor in serum (sTfR), and hepcidin were used, when assessing iron balance in the studied population. It was found that ferritin achieved significantly higher concentrations in the group of men and among smokers. The longer the smoking history expressed in pack-years of smoking, the higher the concentration of hepcidin was. Due to the significant correlation between frailty syndrome and nutrition, the parameters of iron metabolism were also compared with the results of the nutritional scales, body weight, and muscle strength. Higher values of ferritin and hepcidin were observed in people with higher weight loss in the year prior to admission to hospital. The higher MNA score correlated significantly (*p* ˂ 0.05) and positively (r ˃ 0) with iron and TIBC, and significantly (*p* ˂ 0.05) and negatively (r ˂ 0) with ferritin and hepcidin. Higher ferritin values and lower iron and TIBC concentrations were also associated with slower walking speed, which is another essential component of the physical component of frailty syndrome. It has been shown that both frailty syndrome and the associated diagnoses shared a similar pathophysiology and could be characterized by similar deviations in the biochemical profile. Significantly lower iron concentrations were observed in a group of patients with heart failure, renal failure, and anemia. The next stage was the analysis of iron metabolism in the context of patients with frailty syndrome (Table 4). It was shown that iron concentration correlated significantly (*p* ˂ 0.05) and negatively (r ˂ 0) with the total Edmonton Scale score and with such subscales as general health, functional independence, mood, and functional performance (Table 5). TIBC correlated significantly (*p* ˂ 0.05) and negatively (r ˂ 0) with the total score on the Edmonton Scale and with such subscales as cognitive abilities and functional performance (Table 6). No statistical significance was found for UIBC. It has been shown that higher ferritin values occurred in patients who more frequently forgot to take their medications, but did not affect the total Edmonton score. Hepcidin and sTfR concentrations were not found to be related to the combined Edmonton Scale score (Table 7 and Table 8). The analysis of sTfR and hepcidin with CRP concentration was also performed. It was found that hepcidin and sTfR significantly and positively correlated with CRP concentration.

### 3.5. Multivariate Analysis

A multivariate analysis took into consideration the socio-economic status of the studied patients, including such factors as age, gender, residence, and education of the patients. The clinical data for the analysis included the most common diagnoses of comorbidities, muscle strength, nutritional status (including parameters such as albumin, total protein, total cholesterol, high-density lipoprotein (HDL), low-density lipoprotein (LDL), MNA, NRS, waist-hip ratio (WHR), and body mass index (BMI)) and laboratory parameters of inflammation, morphology, and iron metabolism (TIBC, UIBC, iron, hepcidin, sTfR, and ferritin). The female gender has been found to be a risk factor for increased incidence of frailty syndrome. The nutritional status of the studied patients was also considered to be important. Lower BMI and WHR, as well as better nutritional status expressed by higher MNA scores, resulted in a lower frequency of FS. Heart failure was also considered to be a significant risk factor for the occurrence of frailty syndrome. In addition, it was found that there were laboratory parameters related to frailty syndrome. It was found that both an increased concentration of leukocytes and a decreased concentration of iron statistically and significantly increased the frequency of FS in the study group (Table 9).

## 4. Discussion

Frailty syndrome is a significant factor that increases the risk of sudden deterioration of general condition and loss of independence in geriatric patients. The prevalence of frailty syndrome using the Edmonton Frailty Scale in a Portuguese geriatric population was 47.2% and the prevalence was higher in women (48.8%) as compared with men (41.8%). The prevalence of frailty syndrome was even higher in the older age groups (41.3% between 65 and 79 years of age and 65.2% aged 80 or more) [21], which correlated with the frequency of frailty syndrome in the studied group of patients (established at 27.5%) and in a previous study conducted in a Polish population [17]. The factors predisposing the occurrence of frailty syndrome in the studied population included age, lower education, lack of family support, and loneliness [21]. Patients diagnosed with frailty syndrome were also significantly more often hospitalized. Malnutrition and low physical activity were associated with its occurrence [22]. Previous studies [7,23] have found that proinflammatory factors may be directly related to the cascade of activation of the immune system response. In the studied population, the relationship between elevated inflammatory markers and frailty syndrome was proven. Numerous studies [22,23,24,25,26,27] have shown that at least a few elevated cellular and molecular inflammatory mediators were related to developing frailty syndrome. An extremely important element that coexists with frailty syndrome is also anemia, most often anemia of chronic diseases (connected with chronic inflammation of low grade) and deficiency, mainly related to iron deficiency. The aging of populations, especially in Western countries, is increasing the incidence of anemia in the elderly [28]. This study found an association between decreased hemoglobin level and the severity of frailty syndrome. Anemia is mostly due to insufficient iron intake, but infectious diseases and other causes of chronic inflammation can also reduce iron absorption and its availability. Reduced iron availability causes iron-limited erythropoiesis in the bone marrow [29]. Decreased oxygen supply to tissues caused by anemia can lead to weakness, fatigue, and cognitive decline [30]. In fact, among the respondents, a relationship between low hemoglobin concentration and elements of the Edmonton Scale (deterioration of cognitive abilities, as well as functional independence and functional performance) was found. However, studies have shown another difficulty in assigning anemia to the severity of the frailty syndrome. On the one hand, a decrease in the body’s efficiency, related to its hypoxia in the course of anemia, is one of the axial symptoms of the syndrome; on the other hand, it is impossible to assign a single disease to the characteristics of this group of patients. Moreover, a decrease in hemoglobin concentration is one of the characteristics of the natural aging of an organism [30], which, after all, is not synonymous with the presence of frailty syndrome and disability. At the same time, there is no evidence to suggest that changes in iron stores are an inevitable consequence of aging [12]. For this reason, the assessment of iron metabolism has been based on additional parameters such as sTfR and hepcidin. It has been observed that malnutrition, not uncommon in the elderly, may exacerbate the effects of inflammation on iron status biomarkers [31]. Previous observations were also confirmed in this study. People with malnutrition expressed by greater weight loss and higher MNA scores showed significantly lower iron levels and higher ferritin and hepcidin levels. Higher ferritin values and lower iron and TIBC levels also correlated with functional performance as measured by a walking speed test, although there was no correlation between iron metabolism and muscle strength. Interestingly, no significant relationship was found between iron metabolism parameters and age. In the studied population, it was found that patients with frailty syndrome were characterized by a reduced concentration of iron in the blood serum and lower TIBC values. In addition, hepcidin and sTfR concentrations were not found to be related to the combined Edmonton Scale score. The above results lead to the conclusion that a decrease in iron concentration and disturbances in iron metabolism in patients with frailty syndrome are not mainly of a deficient nature and are mostly related to chronic inflammation resulting in the sequestration of iron stores. Similar observations in relation to numerous chronic diseases, mainly associated with the cardiovascular system, have been carried out in the past [32]. While there is a positive association among low serum iron, cardiovascular disease, and all-cause mortality, no unequivocal cause-effect relationship can be inferred [33,34]. The association with anemia is due to physical health, and therefore may primarily reflect anemia of chronic disease [35]. The influence of iron metabolism disorders on the functioning of elderly patients has also been shown in the case of heart failure, which was particularly associated with the occurrence of frailty syndrome in a study [29]. In patients with chronic inflammation, its impact can be particularly severe and can exacerbate the underlying condition, leading to accelerated clinical deterioration [29]. According to the recommendations of Cappelini et al. [34], iron deficiency should be considered to be a separate disease entity and iron deficiency should be defined as a health-related condition in which iron availability is insufficient to meet the body’s needs and which may occur with or without anemia. Regardless of the cause, iron metabolism disorders and iron deficiency are strongly associated with increased hospitalization and death rates [26] as well as the occurrence of frailty syndrome. However, in this study, it was not found that the presence of frailty syndrome was significantly related to hepcidin concentration. The lack of correlation in terms of this biomarker may be associated with an insufficient number of patients and the coexistence of acute inflammatory diseases, also in patients without frailty syndrome, which could significantly affect the statistical correlation. According to the obtained results, it seems that iron metabolism disorder resulting in iron deficiency is strongly associated with the occurrence of frailty syndrome, which may be important in the further prognosis of patients’ quality of life, mortality, and frequency of hospitalization.

## Figures and Tables

**Figure 1 jcm-10-05596-f001:**
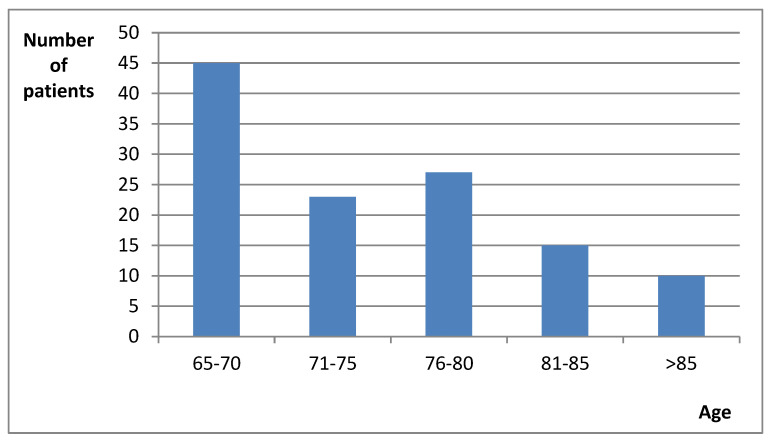
Age structure of patients.

**Table 1 jcm-10-05596-t001:** Causes of hospitalization.

Causes of Hospitalization	*n*
Arterial hypertension	49
Anemia	14
Heart failure	10
Ischemic stroke	7
Pneumoniae	7
Vertigo	4
Fever of unknown origin	3
Diabetes	2
Dyspnea	2
IVIG	2
Thrombocytopenia	2
Syncope	2
Weakness	2
TIA	2
Weight loss	2
Diarrhea	1
Chest pain	1
Polycythemia	1
Coloscopy	1
MDS	1
Atherosclerosis	1
Nephropathy	1
Pancytopenia	1
Faint	1
Cholecystitis	1

IVIG: intravenous immune globulin substitution; TIA: transient ischemic attack; MDS: myelodysplastic syndrome.

**Table 2 jcm-10-05596-t002:** Sociodemographic data.

Parameter		Total Number
Age (years) Men (years) Women (years)	mean ± SD	74.58 ± 7.16
		74.06 ± 7.13
		74.94 ± 7.13
Pack-year *	mean ± SD	14.66 ± 19.16
Gender	Women	71 (59.17%)
	Men	49 (40.83%)
Place of residence	Town >500,000	74 (61.67%)
	Town 100,000–500,000	4 (3.33%)
	Town < 100,000	20 (16.67%)
	Rural area	22 (18.33%)
Smoking (daily use)	No	64 (53.33%)
	Yes	56 (46.67%)
Alcohol (>40 g/week)	No	108 (90.00%)
	Yes	12 (10.00%)
Education	Primary	25 (20.83%)
	Vocational	25 (20.83%)
	Secondary	40 (33.33%)
	Higher	30 (25.00%)

* Pack-year—the number of packs of cigarettes smoked per day multiplied by the number of years the person has smoked.

**Table 3 jcm-10-05596-t003:** The incidence of frailty syndrome in hospitalized patients.

Edmonton Frailty Scale Score	Frailty	*n*
0–4	Non-frail	63
5–6	Risk group	24
7–8	Light frailty	16
9–10	Medium frailty	11
11–17	Severe frailty	6

**Table 4 jcm-10-05596-t004:** Laboratory iron regulation parameters and frailty.

Laboratory Parameters	Non-Frail (0–4 on the EFS)	Pre-Frail (5–6 on the EFS)	Light Frailty (7–8 on the EFS)	Medium Frailty (9–10 on the EFS)	Severe Frailty (11–17 on the EFS)
Iron serum concentration (mg/dL) mean ± SD	82.62 ± 49.72	103.58 ± 71.52	74.94 ± 52.45	73.09 ± 39.57	72.5 ± 65.68
TIBC (umol/L) mean ± SD	304.90 ± 89.5	266.71 ± 138.16	291.56 ± 94	330.27 ± 90.27	312.33 ± 64.6
UIBC (umol/L) mean ± SD	226.86 ± 85.09	217.21 ± 100.65	246.94 ± 84.42	217.27 ± 75.26	239.83 ± 25.15
Hepcidin (ng/mL) mean ± SD	43.33 ± 37.79	58.07 ± 67.9	76.84 ± 55.14	306.42 ± 278.15	44.47 ± 77.49
sTFR (umol/L) mean ± SD	23.15 ± 11.59	26.81 ± 16.7	32.33 ± 19.66	28.79 ± 18.60	22.50 ± 7.77
Ferritin (ug/L) mean ± SD	288.41 ± 793.06	284.13 ± 455.29	296.44 ± 428.47	131.46 ± 104.47	166.95 ± 229.58

EFS—Edmonton Frailty Scale; TIBC—total iron binding capacity; UIBC—unsaturated iron binding capacity; sTfR—soluable transferrin receptor in serum.

**Table 5 jcm-10-05596-t005:** Iron serum concentration and frailty.

Edmonton Frailty Scale	Iron Serum Concentration
	Spearman’s Correlation Coefficient
Edmonton Frailty Scale total score	r = −0.397 **
Cognition score	r = −0.162
General health status score	r = −0.283 *
Functional independence score	r = −0.394 **
Social support score	r = 0.009
Medication use score	r = 0.046
Nutrition score	r = −0.075
Mood score	r = −0.214 *
Continence score	r = −0.152
Functional performance score	r = −0.422 **

* Statistically significant relationship (*p* < 0.05) and ** statistically significant relationship (*p* < 0.001).

**Table 6 jcm-10-05596-t006:** TIBC and frailty.

Edmonton Frailty Scale	TIBC
	Spearman’s Correlation Coefficient
Edmonton Frailty Scale total score	r = −0.199 *
Cognition score	r = −0.321 **
General health status score	r = −0.132
Functional independence score	r = −0.118
Social support score	r = 0.101
Medication use score	r = 0.101
Nutrition score	r = −0.078
Mood score	r = −0.102
Continence score	r = −0.071
Functional performance score	r = −0.24 *

* Statistically significant relationship (*p* < 0.05) and ** statistically significant relationship (*p* < 0.001).

**Table 7 jcm-10-05596-t007:** Hepcidin and frailty.

Edmonton Frailty Scale	Hepcidin
	Spearman’s Correlation Coefficient
Edmonton Frailty Scale total score	r = 0.037
Cognition score	r = 0.101
General health status score	r = −0.029
Functional independence score	r = −0.001
Social support score	r = −0.15
Medication use score	r = −0.222 *
Nutrition score	r = 0.148
Mood score	r = −0.001
Continence score	r = 0.04
Functional performance score	r = 0.099

* Statistically significant relationship (*p* < 0.05).

**Table 8 jcm-10-05596-t008:** sTFR and frailty.

Edmonton Frailty Scale	sTfR
	Spearman’s Correlation Coefficient
Edmonton Frailty Scale total score	r = 0.137
Cognition score	r = −0.178
General health status score	r = 0.1
Functional independence score	r = 0.198 *
Social support score	r = −0.029
Medication use score	r = −0.106
Nutrition score	r = 0.05
Mood score	r = 0
Continence score	r = 0.143
Functional performance score	r = 0.198 *

* Statistically significant relationship (*p* < 0.05).

**Table 9 jcm-10-05596-t009:** Multivariate analysis.

Feature		OR	95% CI		*p*
Age	(years)	0.996	0.812	1.222	0.971
Gender	Women	1	ref.		
	Men	0.006	0	0.229	0.006 *
Place of residence	Rural area	1	ref.		
	Urban area	11.717	0.369	371.563	0.163
Education	Higher	1	ref.		
	Secondary	1.578	0.098	25.333	0.747
	Vocational	1.815	0.077	42.608	0.711
	Primary	3.386	0.216	53.075	0.385
Hand grip, right	(Lb)	1.049	0.872	1.262	0.615
Hand grip, left	(Lb)	0.827	0.602	1.136	0.241
BMI	(kg/m^2^)	0.729	0.56	0.949	0.019 *
WHR	(0,1)	7.729	1.173	50.941	0.034 *
NRS		0.523	0.176	1.557	0.244
MNA		0.605	0.374	0.977	0.04 *
Hospitalization reason	Others	1	ref.		
	Pneumoniae	<0.0001	<0.0001	0.79	0.044 *
	Ischemic stroke	1087.582	0.8	>10000	0.058
	Heart failure	861.387	3.586	>10000	0.016 *
	Anemia	0.005	<0.0001	2.444	0.093
	Arterial hypertension	1.792	0.093	34.661	0.699
CRP	(mg/L)	0.98	0.942	1.019	0.315
Ferritin	(µg/L)	0.999	0.997	1.001	0.254
Iron	(µg/dL)	0.951	0.915	0.989	0.011 *
TIBC	(µg/dL)	0.998	0.976	1.021	0.869
UIBC	(µg/dL)	1.012	0.989	1.036	0.296
Hepcidin	(ng/mL)	1.018	0.983	1.055	0.318
sTfR	(nmol/L)	1.013	0.909	1.129	0.811
LEU	(tys/µL)	1.092	1.007	1.183	0.032 *
HGB	(g/dL)	0.745	0.379	1.465	0.393
PLT	(tys/µL)	0.997	0.986	1.009	0.664
Total protein	(g/dL)	0.096	0.006	1.429	0.089
Albumin	(0.1 g/dL)	0.844	0.512	1.391	0.505
HDL	(mg/dL)	0.93	0.855	1.012	0.093
LDL	(mg/dL)	1.017	0.997	1.037	0.105
Total cholesterol	(mg/dL)	0.987	0.958	1.016	0.362

BMI—body mass index; WHR—waist-hip ratio; NRS—nutritional risc score; MNA—mini nutritional assessement; CRP—C-reactive protein; TIBC—total iron binding capacity; UIBC—unsaturated iron binding capacity; LEU—leukocytes; HGB—hemoglobin; PLT—platelets; HDL—high-density lipoprotein; LDL—low-density lipoprotein. *p*, multivariate logistic regression; * statistically significant relationship (*p* < 0.05).

## Data Availability

The full data presented in this study are available on request from the corresponding author. The data are not publicly available due to patients’ confidentiality.

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
