# Peer review of "Iron Dysregulation and Frailty Syndrome"

_jcm, 2021, doi:10.3390/jcm10235596_

Round 1

Reviewer 1 Report

The manuscript "Iron dysregulation and frailty" from B. Zawadzki et al. tries to establish the relationship between different factors, including iron, and frailty syndrome. The article is well organized and discussed. The references are appropriate and updated.

The suggestions to the authors are:

  1. In the Introduction, the sentence which is between those with reference numbers 8 and 9 has (Eisenstaedt R, 2006). I assume that this should be replaced with the reference number.
  2. In the conclusion of the manuscript authors  say  that according to the obtained results, iron metabolism disorders are a very important risk factors.... Earlier in the discussion authors said (based on the results) that disturbances in iron metabolism are mostly related to chronic inflammation. Therefore, I think that the last sentence in manuscript (conclusion) should be changed in a way that instead "very important risk factor" should be something like  "that iron metabolism disorders influence as a risk factor....).

Author Response

Thank you for your revision and valuable comments. I have shortened the article, especially introduction and discussion so it could  give more focus to the main iron dysregulation topic. I have also corrected information about frequency of frailty in the abstract. I have used the same term 'patients' to describe participants all through.  According to the suggestion of other Review I have changed the presentation of the results as mean +/- SD for groups of patients. Tables 4,5,6,7,8, which have given too much detail and figures 2,3 have been dropped.

I have changed the final sentence in the discussion. 

I am looking forward to hearing form you and further comments. 

Best regards,

B. Zawadzki

Reviewer 2 Report

The general impression of this article is that it has a lack of focus and gives much too much of discussion and presentation of themes of peripheral relevance to the main focus of the article, the association between Frailty syndrome and iron dysregulation.

some further details:

The abstract states that ' ...the frequency of frailty syndrome in the general population is estimated at 17%...'  which is obviously wrong, and refers only to the elderly population (65+?), not to the general population.

Participants in the study are referred to participants, respondents, hosptalized, patients,.., which is confusing. Be consequent and use the same term all through. 

the introduction is much too long and should focus on a brief explanation of what the frailty syndrome is, and why it is of interest to study iron parameters in patients with the syndrome.

The results could better be presented as mean +/- SD for groups of patients according to frailty status, may be:  No frailty, Pre-frailty or light frailty, moderate or servere frailty (or all 5 groups separately). Drop median, quartile, and percentages. Smoking and Alcohol should bebetter explained: Present use? Ever use? Above some limit ?

Tables 4,5,6,7,8 give too much detail and should be dropped.

Figures 2 and 3 is analysis of frailty components and not very relevant for the study of iron regulation, and should be dropped

Tables 9 to 12 should drop the correlations with each component of the Frailty scale and substituted by one Table, something like the following

iron regulation parameter No frailty pre-frail  light moderate severe
Iron serum concentration (mean ,SD) (mean, SD) (mean, SD) (Mean, SD) (mean, SD)
TIBC (mean, SD) . . . .
Hepcidin . . . . .
sTFR . . . . .

In the discussion, which is also too long and spread out on more or less relevant information, associations are presented as risk factors, which they should not.  E.g. is iron dysregulation a cause/risk factor of frailty, or a consequence of same?

Author Response

Thank you for your revision and valuable comments. I have shortened the article, especially introduction and discussion so it could  give more focus to the main iron dysregulation topic. I have also corrected information about frequency of frailty in the abstract. I have used the same term 'patients' to describe participants all through.  According to the suggestion I have changed the presentation of the results as mean +/- SD for groups of patients. Tables 4,5,6,7,8, which have given too much detail and figures 2,3 have been dropped.

So far, I have left the correlations with each component of the Frailty Scale in tables 9-12. I think it is important to show asociations between frailty in general and with particular elements of frailty scale, not only with information about severity of the syndrome. 

I am looking forward to hearing form you and further comments. 

Best regards,

B. Zawadzki

Round 2

Reviewer 2 Report

The revised version of the manuscript is substantially improved.  There are, however, several more points for consideration.

Abstract line 6:  ..affects the...   change to. ..is associated with...

Abastract  line 10: 120 patients....  change to: 120 hospitalized patients >65 years of age participated...

Abstract line 15:  ..patients, the..    Change to: ..patients >65 years of age, the incidence...

Material and methods  lines 8--9: Include a reference for each of the scales

M&M line 12:  MMSE :  explain the acronym

M&M line 20:  median, quartiles, maximum and minimum:  as far as I can see, these results are not used in the presentation and should be omitted here.

Results line 2. Use only one decimal for mean ages

Results lines 3-4:  Delete the percentages; number of patients is sufficient

Figure 1:  The two bar charts are identical and apparently both show the age distribution of the total number of patients.  I guess they are two because the plan was to show age distribution of men and women separately. This must be corrected. Also, there should be a description on each axis: Number of  women/men (so the reader should be told it's not percentages), and 'Age' on the horizontal axis

Table 1: Delete percetages. Number of patients is sufficient (and anyway, there is no sense in giving pecentages with two decimals from a material of 120 patients). 

The text following after Table 1: Here are 5 sentences that belong in the Material and Methods section , but largely already mentioned there (if necessary some specification added in the M&M section, e.g. the classification of cities) and should be omitted here:  The survey assessed...,   The place of residence...,   The survey also investigated...,  The parameters characterizing....,   Regular exercise...  These sentences should be taken out from this paragraph

Table 2: give ages separately for men and women, as in text.  Delete percentages. Explain the term 'Pack-year'

3.2. Edmonton Frailty Scale results: Never start a period with a number ! I suggest reformulating the  three first lines as follows: Sixty-three of the patients (52,5%) were non-frail, 24 (20%) were in the risk group, and 33 (27,5%) had the frailty syndrome. The level of frailty was light in 16, medium in 11, and severe in 6 patients (Table 3).

Table 3: Delete percentages

Previous tables 9-12, now 4-7: The authors find this degree of details to be important, which is acceptable, and the tables can be maintained.  However, I have two suggestions: 

First, to make the tables simpler, I suggest that all the exact p values are dropped, and the r values marked with one asterisk for values significant at the p<0,05 level, and two asterisks for values significant at the p<0.01 level, and the subscript changed accordingly to: * p<0.05; ** p<0.01 

Second, there is a problem with presenting only correlation statistics, since the level of the laboratory values are not revealed, and I believe these would be at least as interesting to the reader. I therefore suggest that a new Table 4 showing mean laboratory values (and SD) for each of the frailty levels (non-, pre/Risk, light, medium, severe), as suggested in my previous report,  is introduced before the present tables 4-7. No significance test is needed for this new table.

New tables 8 and 9. Each of these tables have one single row only. These results should rather be presented in two sentences in the text, not in single-row 'tables'.

Author Response

Thank you for your valuable comments. We took into consideration your revision and changed the manuscript point by point. 

We are looking forward to hearing from you. 

Best regards,

B. Zawadzki